

# Comparison of deep learning models in automatic classification of coffee bean species

Adem Korkmaz[1], Tarık Talan[2], Selahattin Koşunalp[1] and Teodor Iliev[3]

[1] Department of Computer Technologies, Bandırma Onyedi Eylül University, Bandırma, Turkey
[2] Department of Computer Engineering, Gaziantep Islam Science and Technology University, Gaziantep, Turkey
[3] Department of Telecommunications, University of Ruse, Ruse, Bulgaria

## ABSTRACT

As one of the most widely consumed beverages worldwide, coffee is characterized by its diverse flavor profiles and complex production processes. In this study, deep learning-based image processing techniques are employed for the automatic classification of coffee bean species with high accuracy. To achieve this, images of three different coffee bean species (Starbucks Pike Place, Espresso, and Kenya) were classified using five CNN-based models: Xception, DenseNet201, InceptionV3, InceptionResNetV2, and DenseNet121. The dataset comprises 1,554 coffee bean images. Cross-validation was applied to assess the models' performance, and classification accuracy was evaluated using performance metrics. Among the tested models, InceptionV3 achieved the highest classification accuracy (93%) and precision (95%), with the lowest loss rate (0.12), making it the most effective model in this study. As a result of the experiments, the average classification success rates of the models were determined as follows: 93% for InceptionV3, 92% for DenseNet121, 91% for Xception, 91% for InceptionResNetV2, and 90% for DenseNet201. These findings indicate that InceptionV3 demonstrates the highest performance. It is anticipated that this study will make significant contributions to applications in coffee bean classification.

# INTRODUCTION

Coffee is cultivated in more than 70 countries and is prepared by roasting and grinding the beans of tropical trees belonging to the Coffea genus of the Rubiaceae family (*Buonocore, Carratu & Lamberti, 2022*). With an annual global turnover exceeding 10 billion dollars, coffee stands out as the product with the second highest economic value after crude oil (*Sanders, 2019*). This beverage has played a pivotal role in shaping global production and trade relations on a global scale with its economic value and rich nutritional content for centuries. In recent years, interest and demand for coffee have increased significantly and has become a beverage that all segments of society can easily access. The increasing demand for coffee has led to the development of the coffee industry.

Corresponding author
Tarık Talan, tarik.talan@gibtu.edu.tr

This type of beverage has started to become a part of the daily consumption habits of billions of people due to its unique taste and aroma, positive effects on health and energizing effect (*Haile, Bae & Kang, 2020*). Coffee, one of the most consumed beverages after water and tea, is one of the most profitable agricultural products in the world (*Buonocore, Carratu & Lamberti, 2022*). Consumers' coffee preferences vary greatly depending on the taste, aroma, quality and processing method of the coffee, as well as the origin and type of beans (*Do Carmo et al., 2020*; *Unal et al., 2022*). The type of coffee bean has become a critical issue to consider, as it plays an important role in the trade and consumption processes (*Mihailova et al., 2022*). Therefore, correctly identifying the type of coffee beans in particular plays a critical role in ensuring quality control in production processes and determining market prices, as well as in terms of the sustainability of trade by increasing the reliability of the product offered to the consumer (*Hung, Lee & Lin, 2021*). Accurate and effective coffee bean grading is essential to ensure consistency, quality control and standardization in the coffee industry. Often coffee quality is determined by expert judgement, visual inspection or traditional grading methods with limited functionality (*Turi, Abebe & Goro, 2013*). However, traditional sorting methods used in the coffee industry are often time-consuming, inefficient and costly, as well as prone to human error (*Hung, Lee & Lin, 2021*; *Unal et al., 2022*). To address these challenges, improve quality assessment, and achieve high-accuracy classification, automatic detection of coffee beans using digital image processing has emerged as a critical necessity. Deep learning techniques, in particular, offer rapid, non-destructive, and highly effective solutions for image classification, making them a promising approach for automating coffee bean classification. Despite their advantages, research on deep learning applications in coffee bean classification remains limited. There is a clear need for further investigation and integration of these findings into industrial production processes (*Ansar et al., 2021*). In this context, the aim of the research can be listed as follows:

- To compare the performance of different deep learning models in coffee bean classification.
- To identify the most successful model(s) in terms of classification accuracy.
- To evaluate the applicability of deep learning models for automatic coffee bean classification.
- To compare the performance metrics of various deep learning models using coffee bean image datasets.

This study makes a significant contribution to the literature on coffee bean classification. Most existing studies focus on classical machine learning algorithms or basic deep learning models. In contrast, this research provides a comprehensive comparison of both conventional and next-generation deep learning architectures. Specifically, the study employs five state-of-the-art convolutional neural network (CNN) models—Xception, DenseNet201, InceptionV3, InceptionResNetV2, and DenseNet121—to classify coffee beans. The deep learning architectures used in this study were chosen because they provide high accuracy and generalization capabilities in visual classification tasks. The features of

these models, such as deep connections and dense information sharing between layers, allow them to perform strongly in challenging tasks such as classifying visually diverse objects such as coffee beans. The performances of the models were compared and the obtained results were compared with the existing studies in the literature, and a new approach was presented in the classification of coffee bean species. This study provides a significant contribution to the literature in terms of demonstrating the effectiveness of new generation deep learning models that have not been used before for the classification of coffee bean species. In addition, by examining the usability of these methods in industrial applications, it provides a new perspective for the improvement of automatic separation and quality control processes of coffee beans. Therefore, the study can have a significant impact not only in the academic field but also in practical and industrial applications. Comparing the performance of deep learning models and determining the most suitable model is considered an important step towards automating and improving quality control processes in the coffee industry. The main contributions of the study are:

- **Comprehensive comparison** of five advanced CNN models—Xception, DenseNet201, InceptionV3, InceptionResNetV2, and DenseNet121—in coffee bean classification.
- **Application of transfer learning techniques** to improve model performance on limited datasets.
- **Development of an innovative framework** for classifying three distinct coffee bean species (Espresso, Kenya, and Starbucks Pike Place), addressing limitations of prior studies with small and diverse datasets.
- **In-depth analysis of CNN-based models**, exploring their advantages and limitations in coffee bean classification within the context of visual recognition.

These contributions aim to help better understand deep learning methods for visual classification tasks and demonstrate the effectiveness of their applications in the food and agriculture field. The following sections, in turn, will present the related works, the definitions of models, the performance evaluations.

## Related works

In recent years, artificial intelligence technologies have been increasingly utilized in the agricultural sector, leading to a growing body of research in this area. In particular, image processing and machine learning methods play an important role in processes such as the classification and quality control of agricultural products (*Alkanan & Gulzar, 2024*; *Gulzar, 2024*; *Gulzar et al., 2024*). In this context, the use of artificial intelligence-based methods in the classification of coffee beans is increasing. Several recent studies have leveraged advanced technologies and methodologies to automatically detect and classify coffee bean species. However, some of these studies rely on limited datasets and do not sufficiently explore the generalization capabilities of their models. Additionally, many studies fail to fully utilize the potential of deep learning techniques. The current study aims to address these limitations by achieving high classification accuracy using transfer learning methods. Furthermore, by comparing four different deep learning models, an

approach that offers strong generalization capabilities in the classification of coffee types is proposed. This research, which aims to fill these gaps in the literature, contributes to the development of a more powerful and generalizable classification model. As a result, this study discusses the applicability of deep learning-based classification models in a wider range by exploring the possibilities of achieving high accuracy with limited data. This section shares a comprehensive review of the studies conducted on this subject.

*Unal et al. (2022)* classified various coffee bean images (Starbucks Pike Place, Espresso, Kenya) using transfer learning with four different CNN-based models (VGG16, VGG19, SqueezeNet, and InceptionV3). The dataset used in training the models was created specifically for this study. As a result of the analyzes, it was determined that SqueezeNet was the most successful model by achieving the highest accuracy.

*Arwatchananukul et al. (2024)* developed a deep learning based model to classify defects in Thai Arabica green coffee beans. The study found that the MobileNetV3 model achieved superior performance, reaching an accuracy rate of 99.84%. *Thai, Ko & Huh (2024)* developed a real-time application using deep learning technology to automatically classify coffee bean defects. The study shows that compared to traditional visual inspection methods, the deep learning-based system provides speed, accuracy, and efficiency, and provides great improvement in the quality classification of coffee beans. On the other hand, *Boadu et al. (2024)* geographically differentiated African coffee types using handheld NIR spectroscopy and multivariate data processing. The study shows that NIR techniques are an effective tool to determine the geographical origin of coffee types and provide supply chain traceability.

*Mihailova et al. (2022)* used multispectral imaging to quickly and effectively distinguish Arabica and Robusta coffee beans and measure their adulteration levels. The Orthogonal Partial Least Squares Discriminant Analysis model achieved 100% accuracy in classifying the two coffee types and correctly predicted their adulteration levels.

*Gope et al. (2024)* studied the effectiveness of various You Only Look Once (YOLO) models in the detection and classification of green coffee beans. In the study, comparisons between different YOLO versions revealed that especially the customized YOLOv8n model provides high accuracy and precision and this model can be used efficiently in quality control processes in the coffee industry.

In the study of *Turi, Abebe & Goro (2013)*, a computer algorithm was developed to automatically classify coffee beans taken from four coffee growing regions of Ethiopia according to their origin. Automatic classification was performed with an artificial neural network using color, morphology and texture features obtained from 160 images. The results of the study showed that especially the combination of morphological and color features provided high accuracy and revealed that this method can be effective in determining coffee bean quality.

*Chandu, Raveena & Surendran (2024)* investigated the effectiveness of digital image processing and machine learning methods to assess the inherent quality of unprocessed coffee. The study included the processes of obtaining, processing and analyzing images, while artificial neural network (ANN), support vector machine (SVM), and k-nearest neighbors (k-NN) algorithms were used to classify coffee quality based on morphological

and color criteria. In the experiments, it was found that the ANN approach was superior to other methods with an accuracy rate of 89.45%.

*Buonocore, Carratu & Lamberti (2022)* used object detection techniques to automatically classify coffee bean samples according to their types. In their study, the researchers used a dataset of more than 2,500 coffee beans that they created. A CNN based on the YOLO algorithm was used to categorize the coffee beans in the study. The results showed that the YOLO algorithm can accurately distinguish coffee bean types and can be used as an effective method to detect food fraud.

*Adnan et al. (2020)* used UV-Vis and near-infrared (NIR) spectroscopy methods to distinguish Arabica and Robusta green coffee bean types in their studies. UV-Vis spectroscopy provided 97% accuracy by measuring caffeine and chlorogenic acid contents, while NIR spectroscopy achieved 95% accuracy in species discrimination. The results show that both methods are reliable in distinguishing coffee types.

*Hsia, Lee & Lai (2022)* proposed a lightweight deep CNN to detect the quality of green coffee beans. The model includes methods such as rectified Adam, look ahead, and gradient centralization to increase efficiency. The results showed that the model achieved 98.38% accuracy and 98.24% F1 score, providing high accuracy and low computational time.

In the study of *Lee & Jeong (2022)*, the deep coffee dataset containing 1,813 images (the number of defective beans is 434 and the number of normal beans is 1,379) was used. The researchers developed a CNN model to detect two classes of coffee bean images, normal and defective beans. This model achieved an accuracy rate of 90.44% by solving a binary image classification problem.

*Rivalto, Pranowo & Santoso (2020)* used a deep learning-based CNN to distinguish the types of coffee beans and determine the desired quality. In their study, tests were conducted on 617 images of four different types of coffee beans grown in Indonesia. As a result of the test, the system identified objects with an accuracy rate of 74.26%.

*Hendrawan et al. (2021)* conducted a study to classify three Indonesian Arabica coffee bean species, namely Gayo Aceh, Kintamani Bali, and Toraja Tongkonan. For this purpose, AlexNet CNN was used and various optimizer methods such as SGDm, Adam, and RMSProp with learning rates of 0.00005 and 0.0001 were applied for the classification process. In the evaluation conducted on the test data, it was found that the model achieved an accuracy rate of 99.6%.

In their research, *Chang & Huang (2021)* designed a deep learning algorithm to identify defects in coffee beans. The researchers found that pooling layers caused some feature loss or misclassification, and adopted a new dimensionality reduction method to overcome this problem. In experiments conducted on eight types of coffee beans, the overall accuracy rate of the proposed model was determined as 95.2%. The results show that the model works with high accuracy in classifying coffee bean defects.

*Chen, Jhong & Hsia (2022)* developed an approach that combines semi-supervised learning and attention mechanism in the classification of coffee beans. The dataset used in their study contains 2,149 good and 2,468 bad bean images. Experimental results revealed

that the proposed method has high classification performance and achieved an F1 score of 97.21%.

*Febriana et al. (2022)* developed a lightweight and user-friendly smart coffee bean sorting system based on deep learning to classify 8,000 green Arabica coffee beans into four different categories. MobileNetV2 and ResNet-18 models were used to evaluate the performance of this system and achieved an average classification accuracy of 81.31% and 81.12%, respectively.

*Izza & Kusuma (2024)* proposed the evaluation and optimization of a deep learning model using transformer-based architectures to classify green Arabica coffee. The transformer models used in this study include Vision Transformer (ViT), Data Efficient Transformer (DeiT), and Swin Transformer. According to the results of the research, the Swin Transformer model showed the highest classification performance by reaching 84.75%, ViT 82.25%, and DeiT 81.12% accuracy rates on the test data.

*Balakrishnan Jayakumari et al. (2024)* developed a deep learning-based method to classify Robusta coffee beans into nine different classes. In this process, a large dataset of high-resolution images and data augmentation techniques were used. The results showed that the EfficientNet-B0 model was able to distinguish good and bad coffee beans with 100% accuracy even under difficult lighting and background conditions.

*Enriquez et al. (2024)* compared various CNN models (DenseNet, Xception, MobileNet, Inception, ResNet, Inception, VGG16, VGG19, ResNet50, ResNet101, ResNet152) to determine the most effective architecture for coffee bean classification. The results showed that DenseNet performed the best with 98.9% accuracy, followed by MobileNet (98.2%) and ResNet152 (98.0%). VGG19 recorded the lowest accuracy (90.2%) and F1 score (89.9%).

In *Bipin Nair et al. (2023)*, the DenseNet121 model was applied using a dataset of 363 manually classified coffee bean images. The model was analyzed to identify various classifications. The results showed that the model achieved an accuracy rate of 81.89%.

*Yang et al. (2021)* developed a method that combined terahertz (THz) spectroscopy with machine learning methods to determine the geographical origin of coffee beans. In the study, common machine learning techniques such as CNN, linear discriminant analysis (LDA) and SVM were compared to determine the most effective classification model. In order to improve the performance of LDA and SVM models and overcome the dimensionality problem, a smaller and more manageable feature set was created by applying principal component analysis (PCA) and genetic algorithm. The results show that CNN provides the highest performance with 90% accuracy and appropriate variable selection plays a critical role in the model success.

## MATERIALS AND METHODS

This section provides detailed explanations of the dataset, deep learning models, and performance evaluation criteria used in the study. The flow chart of this study is presented in Fig 1.

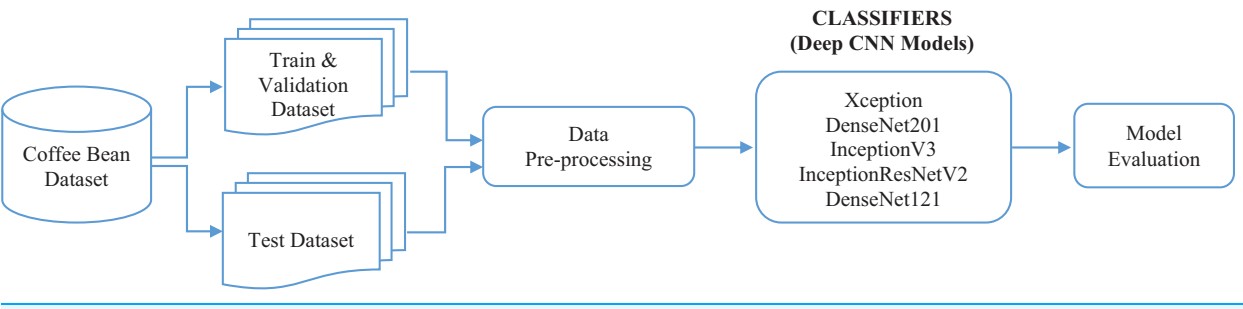

**Figure 1  Flow chart of the proposed study.**               

## Dataset

The dataset used in the present study consists of coffee images obtained from an external source. It is publicly available and was taken from *Koklu*'s *(2024)* website. Detailed information about the dataset, including its collection method, structure, and categorization, can be found in the source by *Unal et al. (2022)*. This dataset includes three types of coffee beans: Espresso (Ethiopia), Kenya (Kenya) and Starbucks Pike Place (Mexico, Costa Rica, Colombia). These types of coffee are both commercially widely used and have different characteristics. Espresso is known as a high-quality coffee and is known for its unique flavor in Kenya. Starbucks Pike Place represents a widespread brand in the global market. It is expected that the classification between these types will contribute to the quality control and type identification processes in the coffee industry (*Unal et al., 2022*). Figure 2 shows the distribution of images in this dataset according to each class.

The dataset comprises a total of 1,554 coffee bean images, consisting of 530 Espresso, 502 Kenya, and 522 Starbucks Pike Place beans.

## Convolutional neural network

CNNs are widely preferred for classifying visual data, such as coffee beans, due to their superior ability to capture spatial relationships within image data. CNN is a deep learning architecture that is frequently used in image processing, natural language processing, and computer vision (*Gu et al., 2018*; *Acikgoz, Korkmaz & Talan, 2024*). CNNs, which are effective in applications such as face recognition, automatic labeling, autonomous vehicles, and intelligent medical treatments, are designed to learn spatial relationships and features in visual data (*Li et al., 2022*).

CNNs typically consist of three primary types of layers. The first is the convolutional layer, which generates feature maps by applying various kernels to images, enabling the model to learn input features. This process facilitates the detection of low-level features such as edges, corners, and patterns. The second is the pooling layer, which reduces the resolution of feature maps to more compact and manageable sizes. To achieve this, techniques such as max pooling or average pooling are commonly used. The final component is the fully connected layer, responsible for high-level reasoning. This layer processes the extracted features to perform tasks such as classification or regression

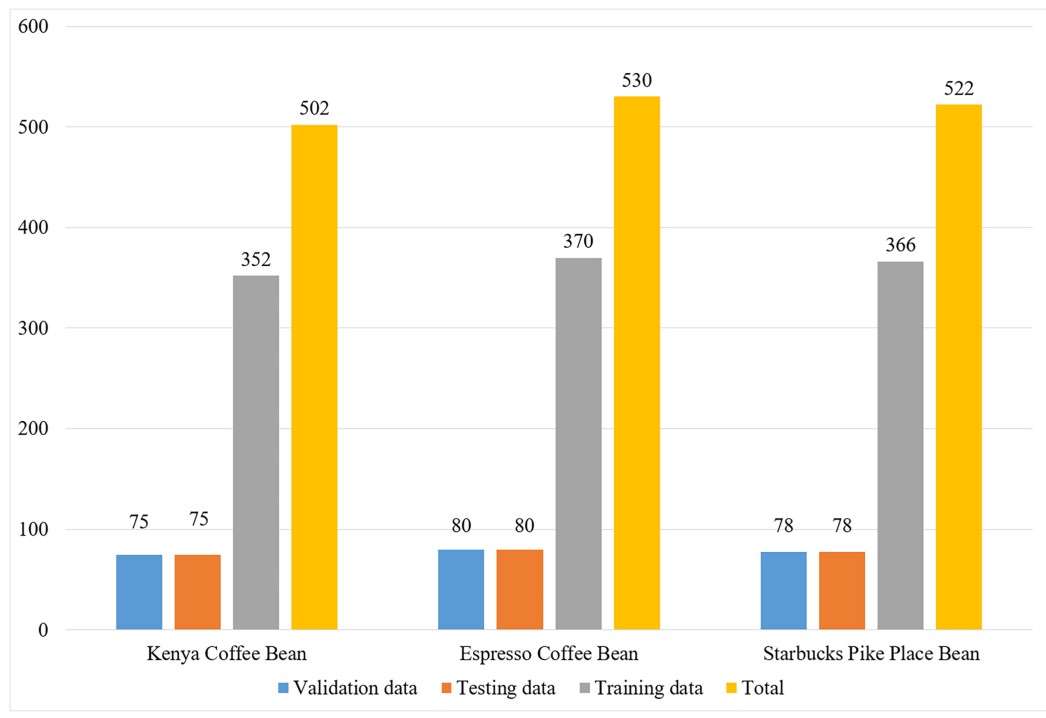

**Figure 2 Distribution of images from the dataset according to classes.**

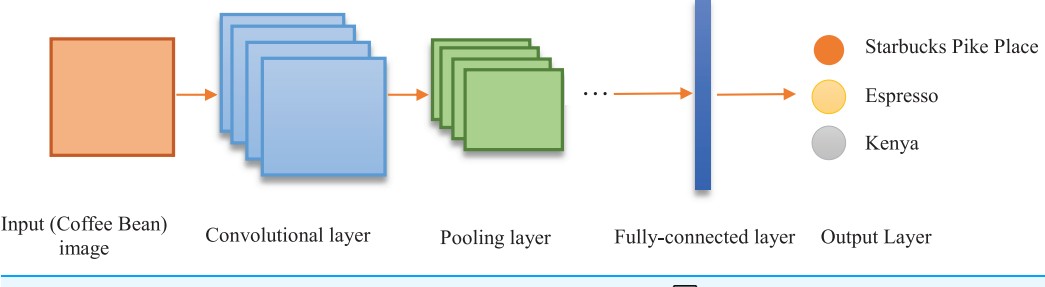

Input (Coffee Bean) image • Convolutional layer • Pooling layer • Fully-connected layer • Output Layer

**Figure 3 Structure of CNN.**

(*Gu et al., 2018*). The integration of these components enhances CNNs' ability to learn complex and hierarchical image features, thereby improving overall performance. The structure of a classical CNN is presented in Fig. 3.

In the study, Xception, DenseNet201, InceptionV3, InceptionResNetV2 and DenseNet121 deep learning architectures were preferred for the classification of coffee beans. Each model was selected due to their strong representation capabilities and high performance that can be achieved with transfer learning. Models such as Xception, InceptionV3 and InceptionResNetV2 offer the ability to learn features at different scales thanks to their deep direct connections and inception structures. This provides a great advantage especially in terms of recognizing the visual diversity and detailed features of coffee beans. DenseNet201 and DenseNet121, on the other hand, provide more efficient information transfer by intensively sharing the information from the previous layers in

each layer and facilitate the learning of deep networks. These structures are known for their strong generalization capabilities and low error rates while increasing the classification accuracy. The different structural advantages of the models, combined with transfer learning, enable us to achieve high accuracy rates in visual recognition tasks such as the classification of coffee beans. This diversity made it possible to obtain more robust and generalizable results by comparing performance between different architectures. These models represent deep learning structures that are frequently preferred for transfer learning. Transfer learning, which is widely used in fields such as speech recognition, natural language processing and computer vision, is a method where a model uses previously learned related knowledge in another similar task more quickly and effectively (*Lu et al., 2015*). This approach is especially useful in cases where the amount of data is limited or the training process takes a long time. In this research, the CNN network was trained using the transfer learning technique and using previously trained models.

### Extreme Inception

The Extreme Inception (Xception) model, introduced by Google in 2017, is designed as an extension of the Inception architecture (*Gülmez, 2023*). This model aims to achieve more efficiency with fewer parameters by using depthwise separable convolution instead of standard convolutions in Inception modules. The model divides a total of 36 convolution layers into 14 blocks. There are linear residual connections in the middle of these blocks in 12 blocks. This structure involves applying independent spatial convolutions to each input channel and then providing inter-channel union with point convolutions (*Chen et al., 2021*; *Gülmez, 2023*). In this way, the model both reduces the computational load and provides an effective solution in deep learning applications by exhibiting high performance on large data sets.

### DenseNet201

DenseNet201, a deep learning model belonging to the dense convolutional network (DenseNet) family, is a modern CNN architecture that exhibits strong performance especially in tasks such as image classification and object recognition. DenseNet201 consists of a total of 201 layers, including an initial layer with a $7 \times 7$ convolution filter and a $3 \times 3$ MaxPool, 196 convolution layers including three average pooling layers, and one output layer (*Bingol, 2022*). DenseNet201 was trained on the ImageNet dataset. This model, which is generally used in complex and multi-class datasets, is built on the dense connectivity principle (*Jesuraj, Evangelin Sonia & Shirley, 2024*; *Salim et al., 2023*). DenseNet201 stands out as an important architecture that increases the efficiency and performance of deep learning models thanks to dense connections.

### InceptionV3

InceptionV3 is a deep learning model developed by Google and introduced in 2015. This model is a version of the Inception architecture and is designed to provide high performance and efficiency in image classification tasks. InceptionV3 has become one of the most accurate models in its field by achieving high accuracy rates on large datasets such

as ImageNet (*Meena, Mohbey & Kumar, 2023*). The model is often used as a pre-trained network for transfer learning and can be re-adapted for various datasets and problems. It can also be applied in areas such as medical image analysis, autonomous driving, security, and surveillance. This model has become an important tool in modern computer vision applications with its multi-scale feature learning, low computational cost, and strong performance. The model has a structure that includes symmetric and asymmetric building blocks, and these structures include components such as convolutions, average pooling, max pooling and fully connected layers (*Chugh et al., 2020*). The Inception-V3 architecture, which has a Softmax function in the last layer, is 48 layers deep and takes an image with a size of 299 × 299 pixels at the input (*Bozkurt, 2021*; *Liu et al., 2021*).

## InceptionResNetV2

InceptionResNetV2 is a deep learning model that combines the flexibility of the Inception architecture and the efficiency of ResNet, offering the advantages of both structures (*Szegedy et al., 2017*). Introduced by Google in 2016, this model was designed to reduce the difficulties encountered in training deep neural networks. The model aims to achieve high accuracy rates, especially in tasks such as image classification. InceptionResNetV2 has 164 layers and approximately 54 million trainable parameters, and can be trained effectively despite its depth thanks to residual connections (*Kanna et al., 2023*). This model prevents gradients from disappearing by adding residual connections to Inception modules and speeds up the learning process (*Kaya, 2024*; *Özgür & Bozkurt Keser, 2021*). As a result, the model shows high performance on large and complex datasets.

## DenseNet121

Proposed in 2016, DenseNet121 is a 121-layer variant of the DenseNet architecture (*Bozkurt, 2021*). DenseNet121 is based on the dense connectivity principle. This allows each layer to establish a direct connection by receiving information from all previous layers (*Huang et al., 2017*). This structural feature helps to prevent the problem of vanishing gradients and increases the efficiency by increasing the reuse of features (*Albelwi, 2022*; *Pleiss et al., 2017*). DenseNet121 offers a balanced performance with a relatively low number of parameters and high accuracy. This model works effectively in deep learning tasks and attracts attention with its high accuracy rates.

## Confusion matrix and evaluation metrics

In this study, a three-class confusion matrix is employed to assess the performance of the classification models. Table 1 presents an example of such a matrix.

In this study, common performance metrics such as accuracy, recall, precision and F-1 score obtained from the confusion matrix are used. In order to calculate these metrics, four basic values are considered in the confusion matrix: true positive (TP), true negative (TN), false positive (FP) and false negative (FN). TP and TN indicate the number of correctly predicted positive and negative examples, while FP and FN indicate the number of incorrectly predicted positive and negative examples. Performance metrics are calculated using the formulas given in Table 2.

**Table 1 Coffee beans confusion matrix.**

| | | Predicted class | | |
|---|---|---|---|---|
| | | Espresso | Kenya | Starbucks Pike Place |
| Actual class | Espresso | $T_1$ | $F_{12}$ | $F_{13}$ |
| | Kenya | $F_{21}$ | $T_2$ | $F_{23}$ |
| | Starbucks Pike Place | $F_{31}$ | $F_{32}$ | $T_3$ |

**Table 2 Formulas for performance metrics.**

| Metrics | Mathematics equation |
|---|---|
| Accuracy | $\dfrac{TP + TN}{TP + TN + FP + FN}$ |
| Recall | $\dfrac{TP}{TP + FN}$ |
| Precision | $\dfrac{TP}{TP + FP}$ |
| F-1 Score | $\dfrac{2TP}{2TP + FP + FN}$ |

*Accuracy*: Measures the proportion of correctly classified samples across all classes. While accuracy is intuitive and useful, it can be misleading in imbalanced datasets where one class dominates, as it may not reflect the model's performance on less represented classes.

*Recall:* Represents the proportion of true positive predictions out of all actual positives in the dataset. Recall is crucial when missing true instances has a high cost, ensuring the model identifies as many true instances as possible.

*Precision:* Indicates the proportion of true positive predictions out of all positive predictions made by the model. Precision is particularly important when the cost of false positives is high, helping avoid misclassifications in those cases.

*F1-score:* The harmonic means of precision and recall, offering a balanced view when both false positives and false negatives have consequences. This metric is ideal for imbalanced datasets where high values of both precision and recall are desired.

## Cross-validation

Cross-validation is a technique used to assess a model's generalization ability and evaluate its performance more reliably (*Göçgün & Onan, 2021*). It is a crucial method for detecting and mitigating issues such as overfitting and underfitting, particularly in complex models like CNNs (*Berrar, 2019*). Cross-validation ensures that the entire dataset is utilized for both training and testing, making it especially beneficial for limited datasets. In this method, the dataset is partitioned into *k* equal subsets. Each subset is sequentially used as the test set, while the remaining *k−1* subsets serve as the training set. This process is repeated *k* times, and the model's overall performance is determined by averaging the results from all iterations (*Unal et al., 2022*).

**Table 3 Details of the parameter settings.**

| Parameters | Value |
| --- | --- |
| Drop-out | 0.3 |
| Optimizer | Adamax |
| Mini-batch size | 16 |
| Input image size | 224 × 224 |
| Learning rate | 0.001 |
| Activation | Relu |
| Loss function | Categorical crossentropy |

## Experiments

This section provides a detailed discussion of experimental and comparative studies. Coffee bean images were classified using five different pre-trained CNN models. The models were trained on a workstation equipped with an Intel® Core™ i7-12700H 2.70 GHz processor, an NVIDIA GeForce RTX 4060 GPU, and 16 GB of RAM. All experiments were conducted using the MATLAB® R2023a software package. Table 3 presents the parameter settings for the models used in this study.

## Data collection and preprocessing

In this study, data collection and preprocessing steps for image classification were meticulously conducted. The dataset used for coffee bean classification was divided into three main subsets: training, validation, and test sets. The training set constitutes 70% of the dataset and was used for the learning process of the model. The validation set covers 15% and was used to tune the model's hyperparameters and monitor the model's accuracy. The test set covers the remaining 15% and was used to evaluate the performance of the final model. These split ratios are a common approach to test the generalization capacity of the model.

Within the scope of the study, firstly, training and test datasets were collected from the relevant folders and organized in a DataFrame structure. File paths and class labels for each image were combined in this structure and brought to a table format. This structure facilitated the processing of images and the organization of the data set. In the data preprocessing stage, all images were scaled to 224 × 224 pixels. This process was done in order to provide data in a suitable size for the input layer of the model. Mini-batch size was determined as 16 during training. Data augmentation techniques were also applied on the training data set in the study. The diversity in the data was increased by adjusting the brightness range on the training data. Data augmentation was used to increase the overall performance of the model and to prevent overfitting. No data augmentation was performed on the test and validation data sets. Thus, the real performance of the model could be evaluated more accurately on these sets. A balanced distribution was provided between the data sets, and distinctions were made carefully for training, validation and test sets. The test set was randomly divided into two and half was used for validation purposes.

**Table 4 Confusion matrix.**

| | Actual class | Predicted class | | |
| --- | --- | --- | --- | --- |
| | | Espresso | Kenya | Starbucks Pike Place |
| Xception | Espresso | 153 | 0 | 7 |
| | Kenya | 0 | 125 | 26 |
| | Starbucks Pike Place | 3 | 4 | 150 |
| DenseNet201 | Espresso | 157 | 0 | 3 |
| | Kenya | 0 | 135 | 16 |
| | Starbucks Pike Place | 17 | 10 | 130 |
| InceptionV3 | Espresso | 156 | 0 | 4 |
| | Kenya | 0 | 141 | 10 |
| | Starbucks Pike Place | 9 | 10 | 138 |
| InceptionResNetV2 | Espresso | 152 | 0 | 8 |
| | Kenya | 0 | 129 | 22 |
| | Starbucks Pike Place | 5 | 6 | 146 |
| DenseNet121 | Espresso | 157 | 0 | 3 |
| | Kenya | 0 | 145 | 6 |
| | Starbucks Pike Place | 11 | 16 | 130 |

This method was preferred to evaluate the performance of the model more accurately and to increase its reliability.

## RESULTS

The networks used for transfer learning were selected among the models trained on ImageNet dataset: Xception, DenseNet201, InceptionV3, InceptionResNetV2 and DenseNet121. In this study, the pre-trained weights of these networks were taken and applied to the classification of coffee beans. The performance of the models was evaluated with accuracy, precision, recall and F-1 score metrics obtained from the confusion matrix of each CNN model. The cross-validation method applied to Xception, DenseNet201, InceptionV3, InceptionResNetV2 and DenseNet121 models analyzed the classification performance between Espresso, Kenya and Starbucks Pike Place coffee bean species. Table 4 presents the performance analysis performed based on the confusion matrix of each model.

The Xception model demonstrated high classification accuracy for the Espresso and Starbucks Pike Place classes, with low misclassification rates. However, it exhibited a higher misclassification rate for the Kenya class compared to the other classes. The DenseNet201 model, on the other hand, achieved the highest accuracy especially in the Espresso class and provided the lowest misclassification rate. This model, which also performed well in the Kenya class, made more misclassifications in the Starbucks Pike Place class, and therefore its performance seems lower than other classes. The InceptionV3 model generally exhibited high accuracy rates in all classes. This model attracted attention by making less misclassifications, especially in the Kenya class, compared to other models.

**Table 5  Performance metrics.**

|  |  | Accuracy | Precision | Recall | F-1 Score |
|---|---|---|---|---|---|
| Xception | Espresso | 0.96 | 0.98 | 0.96 | 0.97 |
|  | Kenya | 0.83 | 0.97 | 0.83 | 0.89 |
|  | Starbucks Pike Place | 0.96 | 0.82 | 0.96 | 0.88 |
| DenseNet201 | Espresso | 0.98 | 0.90 | 0.98 | 0.94 |
|  | Kenya | 0.89 | 0.93 | 0.89 | 0.91 |
|  | Starbucks Pike Place | 0.83 | 0.87 | 0.83 | 0.85 |
| InceptionV3 | Espresso | 0.98 | 0.95 | 0.97 | 0.96 |
|  | Kenya | 0.93 | 0.93 | 0.93 | 0.93 |
|  | Starbucks Pike Place | 0.88 | 0.91 | 0.88 | 0.89 |
| InceptionResNetV2 | Espresso | 0.95 | 0.97 | 0.95 | 0.96 |
|  | Kenya | 0.85 | 0.96 | 0.85 | 0.90 |
|  | Starbucks Pike Place | 0.93 | 0.83 | 0.93 | 0.88 |
| DenseNet121 | Espresso | 0.98 | 0.93 | 0.98 | 0.96 |
|  | Kenya | 0.96 | 0.90 | 0.96 | 0.93 |
|  | Starbucks Pike Place | 0.83 | 0.94 | 0.83 | 0.88 |

In the Starbucks Pike Place class, although its performance was lower than other models, the number of misclassifications was lower. The InceptionResNetV2 model stands out with its low misclassification rates in the Starbucks Pike Place class. However, this model showed a moderate performance in the Espresso and Kenya classes. On the other hand, the DenseNet121 model stands out as one of the models that provide the highest accuracy rates in the Espresso and Kenya classes. However, the correct classification rate of this model is low and the misclassification rate is high in the Starbucks Pike Place class.

In general, the DenseNet201, InceptionV3 and DenseNet121 models showed the best performance and had the lowest misclassification rates in the Espresso class. In the Kenya class, the DenseNet121 model showed the highest performance and had the least misclassifications. In the Starbucks Pike Place class, the Xception and InceptionResNetV2 models showed better performance than the other models and achieved lower misclassification rates.

The high misclassification rate in the Starbucks Pike Place class indicates that this class may have features that are harder to distinguish from others or that there may be more confusion elements for this class in the dataset. These findings help us understand the strengths and weaknesses of the models in certain classes and provide clues about improvements that can be made to improve the classification performance.

Accuracy, precision, recall and F-1 score values of five different models (Xception, DenseNet201, InceptionV3, InceptionResNetV2 and DenseNet121) for Espresso, Kenya and Starbucks classes are shown in Table 5.

The Xception model achieved high performance in the Espresso class, with an accuracy of 0.96, precision of 0.98, and sensitivity of 0.96. The F1-score is 0.97, indicating that the model exhibited a well-balanced performance. In the Kenya class, the accuracy value is

0.83, the precision value is 0.97, and the sensitivity value is 0.83. This shows that the model keeps the rate of false positives low, while making more errors in false negatives. The F-1 score of 0.89 indicates that the model has reasonable performance in this class. In the Starbucks Pike Place class, with high accuracy (0.96) and sensitivity (0.96), but lower precision (0.82), the model generally made correct classifications in this class, but produced false positives to a certain extent. The F-1 score is 0.88. The DenseNet201 model performed well overall, with very high accuracy (0.98) and sensitivity (0.98) in the Espresso class, but slightly lower precision (0.90). The F-1 score was 0.94. In the Kenya class, accuracy was 0.89, precision and sensitivity were 0.93 and 0.89, respectively. This indicates a balanced performance with slightly higher false negatives. The F-1 score was 0.91. In the Starbucks Pike Place class, accuracy was 0.83, precision and sensitivity were 0.87 and 0.83, respectively. The F-1 score was 0.85, indicating that the model performed moderately in this class. The InceptionV3 model performed very strongly in the Espresso class, with very high accuracy (0.98), precision (0.95) and sensitivity (0.97). The F-1 score was 0.96. In the Kenya class, accuracy, precision and sensitivity rates are all 0.93, indicating that the model has a balanced and high performance in this class. The F-1 score is also 0.93. In the Starbucks Pike Place class, accuracy (0.88), precision (0.91) and sensitivity (0.88) rates have shown a good performance in this class. The F-1 score is 0.89. In the Espresso class, the InceptionResNetV2 model has high accuracy (0.95), precision (0.97) and sensitivity (0.95) values, and has generally shown a successful performance. The F-1 score is 0.96. In the Kenya class, accuracy (0.85) and sensitivity (0.85) values are slightly lower than the precision (0.96) value. The model reduced false positives in this class, but produced some false negatives. The F-1 score is 0.90. In the Starbucks Pike Place class, accuracy (0.93) and sensitivity (0.93) are high, while precision (0.83) is slightly lower. The F-1 score is 0.88, indicating good performance in general.

The DenseNet121 model is very successful in the Espresso class with very high accuracy (0.98), sensitivity (0.98) and precision (0.93) values. The F-1 score is 0.96. In the Kenya class, accuracy (0.96), precision (0.90) and sensitivity (0.96) rates indicate that the model is very successful in this class. The F-1 score is 0.93. In the Starbucks Pike Place class, accuracy (0.83), precision (0.94) and sensitivity (0.83) rates indicate that the model performs well in this class, but false negatives are more common. The F-1 score is 0.88.

All models showed high performance in the Espresso class in general. DenseNet201, InceptionV3 and DenseNet121 models were particularly prominent in the Espresso class. In the Kenya class, the InceptionV3 model exhibited the most balanced and highest performance. In the Starbucks Pike Place class, the accuracy rates were generally lower than in the other classes. Xception and InceptionResNetV2 stand out among the models with the highest performance. These differences reveal the capacity of the models to cope with different difficulty levels for each class. The low precision rates in the Starbucks Pike Place class indicate that false positives are more common in this class than in the other classes.

Table 6 presents the training times for the five CNN architectures—Xception, DenseNet201, InceptionV3, InceptionResNetV2, and DenseNet121—measured in minutes. The results highlight notable differences in the computational efficiency of these

**Table 6 Average performance metrics of all models.**

|  | Accuracy | Precision | Recall | F-1 score | MCC | Working hours (min) |
|---|---|---|---|---|---|---|
| Xception | 0.91 | 0.92 | 0.91 | 0.92 | 0.88 | 174.8 |
| DenseNet201 | 0.90 | 0.90 | 0.90 | 0.90 | 0.85 | 317.6 |
| InceptionV3 | 0.93 | 0.93 | 0.93 | 0.93 | 0.89 | 70.4 |
| InceptionResNetV2 | 0.91 | 0.92 | 0.91 | 0.91 | 0.87 | 225.9 |
| DenseNet121 | 0.92 | 0.92 | 0.92 | 0.92 | 0.89 | 157.5 |

models. Among the tested architectures, InceptionV3 demonstrated the shortest training time, completing the process in 70.4 min, which reflects its lightweight design and efficient architecture. This is a significant advantage, particularly for real-time or resource-constrained applications.

On the other hand, DenseNet201 required the longest training time at 317.6 min, reflecting the model's complexity and the increased number of parameters that need optimization. Similarly, InceptionResNetV2 and Xception exhibited longer training times of 225.9 min and 174.8 min, respectively, indicating their computationally intensive nature compared to InceptionV3. The DenseNet121 model, with a training time of 157.5 min, balances computational efficiency and model complexity, making it a viable alternative for scenarios requiring moderate computational resources.

These findings underscore the trade-off between computational cost and performance, as observed in the study. While InceptionV3 provides the most computationally efficient solution with excellent classification performance (93% accuracy), models such as DenseNet201 and InceptionResNetV2 require significantly more training time. This highlights the importance of selecting models based on the specific needs of the application, balancing computational efficiency with accuracy and other performance metrics.

In conclusion, the comparison of training times reaffirms InceptionV3 as the most efficient and practical model for real-world applications, particularly when computational resources are limited. However, for use cases where high resource availability is not a constraint, models like DenseNet201 and InceptionResNetV2 may be explored for their potential advantages in handling more complex feature representations.

The comparison of performance metrics of all classification models used in the study is presented in Fig. 4.

Figure 4 provides a comprehensive comparison of the performance of five deep learning models—Xception, DenseNet201, InceptionV3, InceptionResNetV2, and DenseNet121—across key metrics, including accuracy, precision, recall, F1-score, and MCC. Among these, InceptionV3 demonstrates the highest performance, achieving the best scores across all metrics, with an accuracy, precision, recall, and F1-Score of 93%, and an MCC of 89%. DenseNet121 closely follows, also achieving an MCC of 89%, making it a competitive option for applications requiring high accuracy with lower computational costs. Xception and InceptionResNetV2 exhibit slightly lower but consistent performance across all

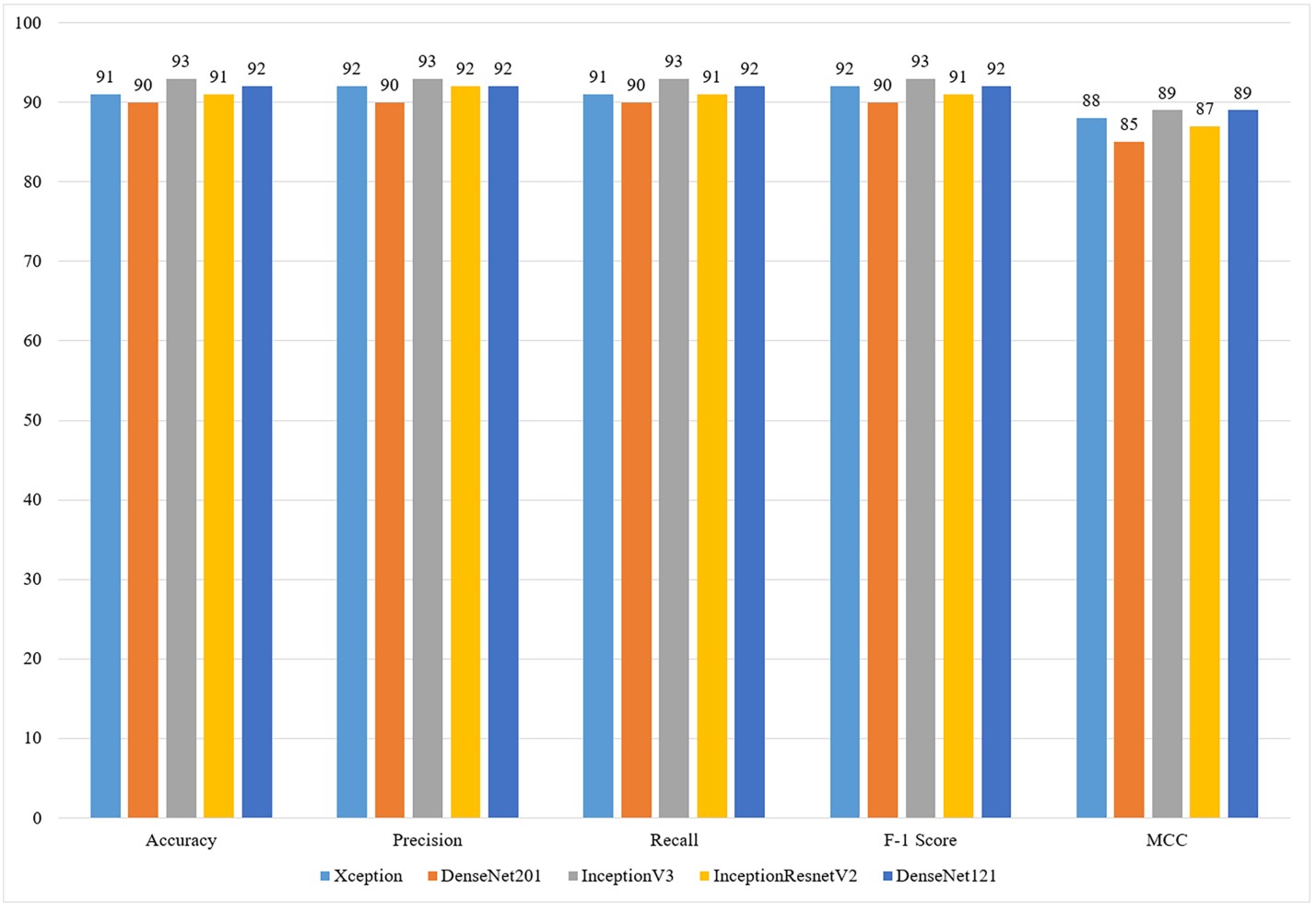

**Figure 4 Comparison of the performances of the models (%).**

metrics, while DenseNet201 shows relatively weaker performance, particularly in MCC (85%), suggesting it may be less effective in handling class imbalances. These results highlight InceptionV3 as the most reliable model for coffee bean classification while showcasing DenseNet121 as an efficient alternative for resource-constrained applications. The figure underscores the importance of balancing classification accuracy and computational efficiency in selecting the optimal model for specific use cases.

ROC curves allow you to evaluate the discriminative properties of models. Figure 5 shows the ROC curves of the models used.

Table 7 presents the pairwise comparisons of the classification models using McNemar's test to assess the statistical significance of their performance differences. The test was conducted by analyzing discrepancies in predictions across all classes (Espresso, Kenya, and Starbucks Pike Place). For each model pair, the test statistic and $p$-value were calculated, as shown in the table. The results indicate that none of the pairwise comparisons yielded statistically significant differences in performance, as all $p$-values exceed the commonly used significance threshold ($p < 0.05$). For instance, the comparison

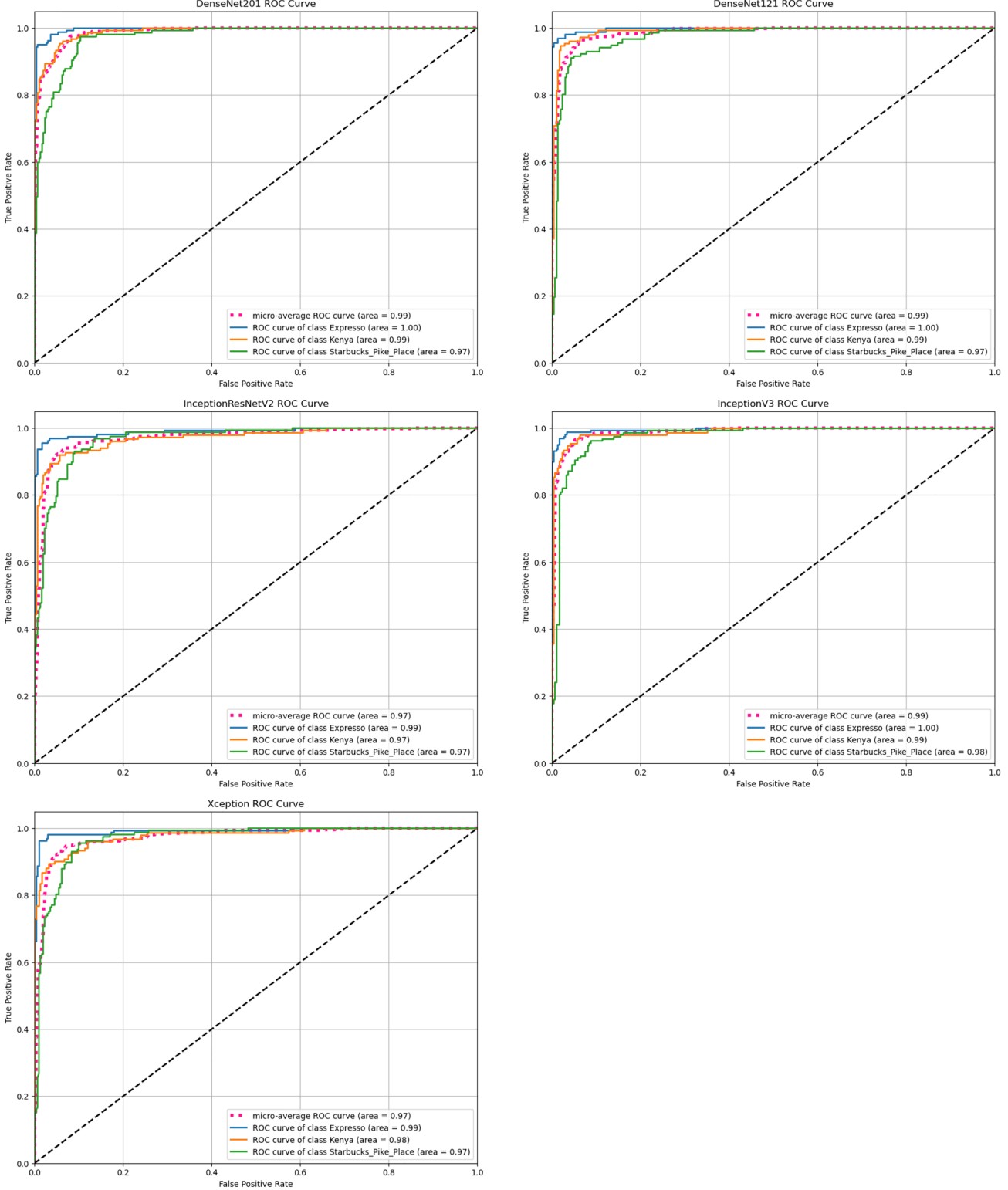

**Figure 5  ROC curves of all models.**

**Table 7 McNemar's test results for model comparisons.**

| Model 1 | Model 2 | Statistic | *p*-value |
|---|---|---|---|
| Xception | DenseNet201 | 0 | 1 |
| Xception | InceptionV3 | 0 | 1 |
| Xception | InceptionResNetV2 | 0 | 1 |
| Xception | DenseNet121 | 0 | 1 |
| DenseNet201 | InceptionV3 | 0.166666667 | 0.683091398 |
| DenseNet201 | InceptionResNetV2 | 0 | 1 |
| DenseNet201 | DenseNet121 | 0.25 | 0.617075077 |
| InceptionV3 | InceptionResNetV2 | 0 | 1 |
| InceptionV3 | DenseNet121 | 0 | 1 |
| InceptionResNetV2 | DenseNet121 | 0 | 1 |

between InceptionV3 and DenseNet121, two of the top-performing models, resulted in a *p*-value of 1.000, demonstrating that the observed differences in accuracy and other metrics between these models are not statistically significant. Similarly, comparisons involving Xception and other models, such as DenseNet201 and InceptionResNetV2, also revealed no significant differences, with *p*-values consistently above 0.05.

The lack of statistical significance across all comparisons suggests that while the models exhibit slight numerical variations in performance metrics (*e.g.*, accuracy, precision, recall), these differences are not large enough to conclusively favor one model over another. This finding aligns with the overall trend observed in the evaluation metrics, where all models demonstrated strong and comparable classification capabilities.

From a practical standpoint, the absence of significant differences implies that the choice of model can be influenced by other factors, such as computational efficiency, resource constraints, or ease of deployment, rather than purely numerical performance. For example, InceptionV3, with its highest accuracy (93%), may still be the preferred choice for applications prioritizing classification performance, whereas DenseNet121, with its computational efficiency, might be better suited for resource-limited environments.

Overall, Table 7 reinforces the robustness of all tested models and highlights the importance of considering additional factors beyond performance metrics when selecting a model for real-world applications. Future studies with larger datasets or additional statistical analyses may provide further insights into potential performance distinctions. Table 8 presents the comparison of experimental studies.

The comparison table illustrates the varying degrees of success achieved by different machine learning models in the classification of coffee beans. EfficientNet-B0, as demonstrated by *Balakrishnan Jayakumari et al. (2024)*, achieved a remarkable accuracy of 100%, suggesting that even lightweight models can excel when applied to smaller, well-structured datasets. However, in contrast, models such as *Rivalto, Pranowo & Santoso (2020)*, which reported an accuracy of 74.26%, highlight the challenges of using smaller datasets and less sophisticated CNN architectures.

**Table 8 Comparison of experimental studies.**

| Author(s) | Dataset | Models used | Accuracy (%) |
|---|---|---|---|
| *Unal et al. (2022)* | 1,554 coffee bean images | VGG16, VGG19, SqueezeNet, Inception V3 | SqueezeNet: 87.3% |
| *Buonocore, Carratu & Lamberti (2022)* | 2,500 coffee beans | YOLO | 94% |
| *Adnan et al. (2020)* | UV-Vis and NIR spectroscopy data | UV-Vis and NIR Spectroscopy | UV-Vis: 97% |
| *Hsia, Lee & Lai (2022)* | 4,626 coffee bean images | LDCNN | 98.38% |
| *Lee & Jeong (2022)* | Deep coffee dataset (1,813 images) | CNN | 90.44% |
| *Rivalto, Pranowo & Santoso (2020)* | 617 images of coffee beans | CNN | 74.26% |
| *Hendrawan et al. (2021)* | 500 images of coffee beans | AlexNet CNN | 99.6% |
| *Chang & Huang (2021)* | 7,203 images of coffee beans | AlexNet, VGG16, GoogLeNet and resNet | AlexNet: 95.2% |
| *Chen, Jhong & Hsia (2022)* | 4,617 coffee bean images | NFNet-F3 | 97.21% (F1 score) |
| *Febriana et al. (2022)* | 8,000 green Arabica beans | MobileNetV2, ResNet-18 | MobileNetV2: 81.31% |
| *Izza & Kusuma (2024)* | 8,000 green Arabica beans | ViT, DeiT, Swin Transformer | Swin Transformer: 84.75% |
| *Balakrishnan Jayakumari et al. (2024)* | 464 coffee bean images | ResNet-34, VGG-16, Inception v3, and EfficientNet-B0 | EfficientNet-B0: 100% |
| *Enriquez et al. (2024)* | 5,000 coffee bean images | DenseNet, Xception, MobileNet, InceptionResNet, VGG16, ResNet50, ResNet152 | DenseNet: 98.9%, |
| *Bipin Nair et al. (2023)* | 363 manually classified coffee bean images | DenseNet121 | 81.89% |
| *Yang et al. (2021)* | 96 coffee bean images | CNN, LDA, SVM with PCA and genetic algorithm | CNN: 90% |
| Our study | 1,554 coffee bean images | Xception, DenseNet201, InceptionV3, InceptionResNetV2 and DenseNet121 | InceptionV3: 93% |

Our study stands out in this comparative analysis, especially with InceptionV3, which achieved an impressive accuracy of 93%. This performance places our model among the top-performing architectures, rivaling those used in other studies, such as *Buonocore, Carratu & Lamberti (2022)* with YOLO (94%) and *Hsia, Lee & Lai (2022)* with LDCNN (98.38%). The strength of our study lies in the successful application of transfer learning using pre-trained models on the ImageNet dataset, which has proven to be highly effective in tackling the complex task of coffee bean classification across multiple classes. Moreover, DenseNet121 and Xception, which also exhibited strong performance in our study, further validate the robustness of our model selection.

Another important observation is the application of transformer-based models, such as the Swin Transformer used by *Izza & Kusuma (2024)*, which achieved an accuracy of 84.75%. Although transformer architectures are increasingly being adopted in image classification tasks, our findings suggest that more traditional CNNs like InceptionV3 and DenseNet201 still hold a competitive edge, particularly in complex, multi-class classification problems such as ours.

The significance of our study becomes more apparent when considering the broader context of coffee bean classification research. The diversity of models and datasets in the literature highlights the challenges inherent in achieving high accuracy, particularly when classifying multiple coffee bean types. Our study not only demonstrates the efficacy of cutting-edge CNN architectures but also offers valuable insights into the classification of challenging datasets, such as Starbucks Pike Place, which presented notable classification difficulties across various studies.

In conclusion, our study contributes meaningfully to the field by demonstrating the high potential of advanced CNN models like InceptionV3 and DenseNet121 in achieving robust classification accuracy, even in challenging multi-class scenarios. These further cements the importance of model selection and dataset management in the pursuit of optimal classification performance. The findings from our study align with the broader trends observed in existing literature yet underscore the unique contributions our approach makes in pushing the boundaries of coffee bean classification research.

## DISCUSSION

This study explores the classification of three different coffee bean species—Starbucks Pike Place, Espresso, and Kenya—using deep learning models trained with transfer learning. The results highlight the significant impact of model selection on classification performance, particularly in the context of visually similar coffee species.

The InceptionV3 model achieved the highest classification success rate, with an overall accuracy of 93%. This model outperformed DenseNet121 (92%) and other models such as Xception and InceptionResNetV2 (both at 91%). InceptionV3 demonstrated superior performance across multiple evaluation metrics, including accuracy, precision, recall, F-1 score, and MCC. Furthermore, the ROC curve analysis reinforced its excellent discriminative ability, making InceptionV3 the most reliable model for coffee bean classification in this study.

One of the most challenging aspects of this study was the classification of Espresso and Starbucks Pike Place beans, which exhibited considerable visual similarities. As evidenced by the confusion matrix, these two coffee species were more frequently misclassified compared to Kenya beans. This challenge is likely due to overlapping visual features, such as similar color tones, textures, and bean sizes, which may have created ambiguity in the models' learned feature representations. Additionally, variations in lighting conditions, bean orientations, or subtle intra-class differences may have further contributed to these misclassifications. These findings highlight a common limitation in visual-based classification tasks, where models relying solely on image features may struggle to differentiate between classes with high visual resemblance. Addressing this issue could involve incorporating non-visual features, such as chemical composition data from spectroscopy, to complement image data and enhance feature differentiation. Furthermore, advanced data augmentation techniques, such as domain-specific transformations, could increase training data diversity, enabling the models to better generalize to subtle inter-class variations. These approaches, combined with further

refinement of model architectures, could significantly improve classification accuracy for visually similar coffee types and overcome the challenges observed in this study.

Another notable finding is the overall effectiveness of CNN-based models, particularly when trained *via* transfer learning. All five models tested in the study provided strong classification results, with relatively low misclassification rates. This demonstrates the potential of deep learning architectures to handle complex classification tasks in the coffee industry. The success of these models suggests that combining CNN architectures with accessible commercial digital scanners could pave the way for more efficient and cost-effective methods of coffee bean classification. These holds promise for non-destructive, real-time identification of coffee bean species, benefiting both producers and consumers.

This study highlights the potential of artificial intelligence technologies in agriculture by examining the use of deep learning methods in the automatic classification of coffee bean species. The findings revealed the performance differences of various model architectures and identified the strengths of each model in terms of classification accuracy. However, transferring these findings to real-world applications could significantly increase efficiency and quality in the sector. Automatic classification of coffee bean species can minimize human errors and increase overall efficiency in the sector, especially by speeding up quality control processes. For example, such systems can increase final product quality by allowing coffee producers to quickly distinguish and classify high-quality beans according to quality. Furthermore, considering the scalability of these systems, high accuracy can be achieved at lower costs by digitizing quality control processes in agriculture. However, there are some challenges to the successful application of this model in practice. In particular, the generalizability and accuracy of the model should be increased under different environmental conditions and for different coffee bean species. In order to be applied to wider areas, it is necessary to test it with datasets from different geographical regions and to evaluate the sensitivity of the model to environmental factors. In addition, by integrating automatic classification systems, it will be possible for small-scale producers in the sector to benefit from these technologies. In future studies, testing this model under various conditions and training it with larger, more diverse data sets will further strengthen its success in the field of application. Thus, quality control processes in coffee production can be made more efficient and competitive advantage can be achieved in international markets.

An important consideration for future development involves the integration of transformer-based approaches, such as Vision Transformer (ViT) and Swin Transformer, which have shown promise in image classification tasks through their ability to capture global features. While these models may require larger datasets and higher computational resources compared to CNNs, they offer scalability and adaptability for complex classification problems. Incorporating these advanced architectures alongside CNNs could potentially enhance model performance, especially for visually similar classes like Espresso and Starbucks Pike Place, where traditional CNNs may encounter challenges. Additionally, comparing the efficiency and accuracy trade-offs between CNNs and transformer-based models under real-world conditions could provide deeper insights into their practical applications. These advancements would further solidify the role of artificial intelligence in

modernizing agricultural processes, ensuring the scalability and robustness of automatic classification systems for diverse coffee bean types and global production scenarios.

## CONCLUSIONS

The results of this study underscore the significant potential of deep learning models for the automatic classification of coffee bean species. By evaluating five popular CNN architectures—Xception, DenseNet201, InceptionV3, InceptionResNetV2, and DenseNet121—this research identified InceptionV3 as the top-performing model, achieving an average accuracy of 93%. The model consistently outperformed others across all major metrics, including precision, recall, and F-1 score, while demonstrating superior discriminative capabilities in ROC curve analysis.

A pivotal contribution of this study is the statistical validation of model performance through McNemar's test, as detailed in Table 7. Although InceptionV3 exhibited the highest numerical performance across key metrics, the test results revealed no statistically significant differences between its performance and those of DenseNet121 (92%) and Xception (91%). This finding highlights the robustness of all tested CNN architectures for classifying coffee bean species. Furthermore, the absence of significant differences emphasizes the importance of practical considerations, such as computational efficiency and ease of deployment, in selecting models for real-world applications.

An important consideration in this study is the computational efficiency of the tested models, as summarized in Table 6. Among the five CNN architectures, InceptionV3 demonstrated the shortest training time (70.4 min) while achieving the highest classification performance (93% accuracy), highlighting its suitability for real-world applications requiring both efficiency and accuracy. In contrast, models like DenseNet201 (317.6 min) and InceptionResNetV2 (225.9 min) required significantly longer training times, reflecting their higher computational complexity. These findings underscore the importance of balancing computational cost with performance metrics when selecting models for practical use. For applications with limited computational resources, InceptionV3 offers a highly efficient solution, whereas more complex models may be preferred in scenarios where resource constraints are not a concern and additional performance gains are sought. This analysis further emphasizes the need to consider both accuracy and computational efficiency when optimizing classification systems for scalable and practical deployment in the coffee industry.

An additional insight is the challenge posed by visually similar coffee bean types, such as Espresso and Starbucks Pike Place, which resulted in higher misclassification rates. This underscores the need for further refinement in model architectures and the incorporation of additional features, such as chemical composition data, to improve classification accuracy. Addressing these challenges would enable more precise differentiation between visually similar types, ultimately enhancing the practical applicability of the models.

The broader implications of this research lie in its potential to transform quality control processes within the coffee industry. The study demonstrates that CNN models, particularly InceptionV3, can provide a non-destructive, cost-effective, and accurate method for real-time coffee bean classification. This advancement could minimize human

errors, improve operational efficiency, and ensure the delivery of more reliable and consistent products to consumers and producers. Moreover, accurate classification can play a critical role in combating food fraud by ensuring authenticity and maintaining product integrity throughout the supply chain.

To further enhance the predictive power and applicability of these models, future work should focus on expanding the dataset with a larger variety of images, representing diverse coffee types and geographical regions. This would improve model generalizability and classification accuracy. Additionally, the integration of advanced architectures, such as transformers or hybrid models, could offer enhanced performance, particularly in differentiating visually similar coffee types. By addressing these areas, future research could further optimize classification systems and contribute to more scalable, robust, and innovative solutions for quality control in the coffee industry.

In conclusion, this study highlights the strong potential of deep learning models for coffee bean classification and lays a foundation for future research aimed at optimizing classification methodologies and advancing agricultural quality control practices.

### Limitations

This study applied five prominent deep learning algorithms—Xception, DenseNet201, InceptionV3, InceptionResNetV2, and DenseNet121—to classify three distinct coffee bean species (Espresso, Kenya, and Starbucks Pike Place) using the dataset provided by *Unal et al. (2022)*. While the dataset facilitated the achievement of successful results aligned with the study's scope and objectives, it is limited in both sample size and diversity. This limitation may constrain the generalization capability of the models, particularly when applied to datasets with broader variability or differing characteristics.

The restricted diversity of the dataset poses challenges in evaluating the robustness of deep learning models across a wider range of real-world scenarios. In particular, the dataset's focus on specific market segments might hinder the models' ability to perform well on coffee bean species from different geographical regions or with distinct visual features. As such, the validity of the findings needs to be tested on more comprehensive datasets that include a greater variety of coffee bean images and species.

Future studies should aim to address these limitations by employing larger, more diverse datasets that encompass a wide range of coffee species, origins, and processing methods. Expanding the dataset in this way would enhance the models' predictive power, improve their generalization to unseen data, and provide more robust insights into the application of deep learning in the coffee industry. By addressing these challenges, future research can further refine the methodologies and extend the practical applicability of these models.

## ACKNOWLEDGEMENTS

During the preparation of this manuscript the authors used ChatGPT-4 in order to improve language and readability.

### Funding

This study is financed by the European Union-NextGenerationEU, through the National Recovery and Resilience Plan of the Republic of Bulgaria, project no. BG-RRP-2.013-0001-C01. There was no additional external funding received for this study. The funders had no role in study design, data collection and analysis, decision to publish, or preparation of the manuscript.

### Grant Disclosures

The following grant information was disclosed by the authors:
European Union-NextGenerationEU: BG-RRP-2.013-0001-C01.

### Competing Interests

The authors declare that they have no competing interests.

### Author Contributions

- Adem Korkmaz conceived and designed the experiments, analyzed the data, performed the computation work, prepared figures and/or tables, and approved the final draft.
- Tarık Talan conceived and designed the experiments, performed the experiments, analyzed the data, performed the computation work, prepared figures and/or tables, and approved the final draft.
- Selahattin Koşunalp performed the experiments, performed the computation work, authored or reviewed drafts of the article, and approved the final draft.
- Teodor Iliev analyzed the data, performed the computation work, authored or reviewed drafts of the article, and approved the final draft.

### Data Availability

The code is available in the Supplemental File.

The Coffee Image Dataset used in the study is available at Kaggle: https://www.kaggle.com/datasets/tariktalan/coffee-image-dataset.

### Supplemental Information

Supplemental information for this article can be found online at http://dx.doi.org/10.7717/peerj-cs.2759#supplemental-information.

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
