# Peer review of "Comparison of deep learning models in automatic classification of coffee bean species"

_PeerJ Computer Science, doi:10.7717/peerj-cs.2759_

## Round 0.1 · original submission · Major Revisions

· Academic Editor

Major Revisions

After a thorough evaluation by two expert reviewers, we have determined that the manuscript requires major revisions before it can be reconsidered for publication. Below, I summarize the main observations and areas for improvement to meet the journal’s quality standards.

Both reviewers agreed that the manuscript addresses a relevant topic in the field of artificial intelligence applied to agriculture, focusing on the automatic classification of coffee bean species. However, the current writing quality needs substantial improvement to enhance clarity, professionalism, and grammatical accuracy. Specific sections, such as the abstract and introduction, require restructuring to ensure coherence and an appropriate academic tone.

Regarding the literature review, the reviewers noted that it does not adequately reflect the most recent advancements in the field, with most references predating 2023. It is crucial to include and critically discuss recent studies to better position the contribution of this work within the current research landscape. Additionally, references must adhere to the journal’s formatting guidelines.

Methodological aspects and the presentation of results were also identified as requiring significant enhancement. Greater detail is needed to justify the selection of the dataset and the specific coffee bean classes included in the study. A more comprehensive description of preprocessing steps, data augmentation techniques, and model configurations is also necessary. Furthermore, the presentation of results would benefit from the addition of graphical elements, such as confusion matrices and training-validation curves, as well as statistical tests to validate the differences in model performance.

The reviewers emphasized the need for substantial improvements in the quality and clarity of the figures, particularly Figures 2 and 4. Tables should be updated to include more detailed comparisons with state-of-the-art techniques, further enriching the manuscript.

The discussion section should expand on the practical implications of the study, highlighting how the findings can be applied to real-world scenarios in coffee bean classification. Additionally, a more in-depth analysis of the trade-offs between accuracy and computational efficiency, as well as the limitations of the proposed models, is necessary.

We require a detailed point-by-point response to the reviewers’ comments in a response letter, explaining how each concern has been addressed. Please also highlight all changes made in the revised manuscript to facilitate the review process.

We value the potential of this work to advance the knowledge and application of machine learning in agriculture and look forward to receiving a revised version that thoroughly addresses the feedback provided.

Reviewer 1 ·

Basic reporting

• Language and Writing Quality: Writing is not professionally up to the level for Example in the Abstract: “Coffee stands out as one of the most widely consumed beverages around the world” does not look in standard. (3/5)
• Article Structure: yes (4/5)
• Literature Review: References used are very old (2013, 2021, 2022), only few papers they have discussed about 2023, and 2024. Literature review is not in trend. Reference also not in journal format. Check it “Smith JL, Jones P, Wang X. 2004. Investigating ecological destruction in the Amazon. Journal of the Amazon Rainforest 112:368-374 DOI: 10.1234/amazon.15886.”
• Figures and Tables: Performance metrics need enhancement Figure 1 illustrates a fundamental representation of Convolutional Neural Networks. Figure 2 displays unclear photos of coffee beans. Require enhancement of the picture quality. The architectural diagram in Figure 4 requires improvement. Tables are rudimentary and need comparison with more advanced deep learning techniques.
• Data Availability: Dataset discussed is very less, Expresso, Kenya, and Starbuck pike place. For example, primary source coffee beans have (Typica, Bourbon, Caturra, SL28 and SL34, Geisha (or Gesha), Pacamara, Maragogipe (Elephant Bean).

Overall Feedback:
"The article is well-written and adheres to the journal's structure. However, the introduction lacks recent references on [2024], and Figure 2, Figure 3, and Figure 4 needs a more detailed caption for clarity." Dataset is less.
Author can check for reference cite for improving the paper
 Gope, H.L., Fukai, H., Ruhad, F.M. and Barman, S., 2024. Comparative analysis of YOLO models for green coffee bean detection and defect classification. Scientific Reports, 14(1), p.28946.
 Raveena, S. and Surendran, R., 2023, January. ResNet50-based classification of coffee cherry maturity using deep-CNN. In 2023 5th International Conference on Smart Systems and Inventive Technology (ICSSIT) (pp. 1275-1281). IEEE.
 Gope, H.L., Fukai, H., Ruhad, F.M. and Barman, S., 2024. Comparative analysis of YOLO models for green coffee bean detection and defect classification. Scientific Reports, 14(1), p.28946.
 Chandu, B., Raveena, S. and Surendran, R., 2024, October. Upgrading Coffee Bean Quality Using K-nearest Algorithm over Future Selection and Extraction to Reduce Dimensionality of Data. In 2024 8th International Conference on I-SMAC (IoT in Social, Mobile, Analytics and Cloud)(I-SMAC) (pp. 938-943). IEEE.
 Selvanarayanan, R., Rajandran, S. and Alotaibi, Y., 2023. Using Hierarchical Agglomerative Clustering in E-Nose for Coffee Aroma Profiling: Identification, Quantification, and Disease Detection. Instrumentation, Mesures, Métrologies, 22(4).

Experimental design

• Research Design and Objectives: yes, the objectives of comparing deep learning models clearly stated but, experimental setup is basic for achieving these objectives. Author just have compared with pre trained model that shows, no novelty in the paper.
• Details of Implementation: The datasets used in training and testing clearly described but, how it is implemented in real world environment is not explained clearly. the preprocessing steps, data augmentation, and model configurations lack of explanation. Need to work more in this part. Results part lack with less graphical outputs.
OverallFeedback:
"The authors provide a good overview of the dataset used; however, the lack of information on how the data were split into training, validation, and test sets limits the reproducibility of the experiment. Further, the rationale for choosing the specific models (e.g., CNN, Transformer) should be elaborated to justify their relevance to coffee bean classification."

Validity of the findings

• Performance Metrics: yes, the evaluation metrics (e.g., accuracy, F1-score, confusion matrix) appropriate for classification tasks but, overall comparison graph figure 5, is not explained clearly. Only Roc curves for statistical analyses conducted to validate the findings. More research should be done in statistical analyses part.
• Limitations of the proposed model is not explained clearly
Xception: While Xception shows high precision and recall, its F1-Score is slightly lower. What are your thoughts on this discrepancy?
DenseNet201: This model exhibits relatively balanced performance across all metrics. Do you believe its simplicity makes it a strong contender?
InceptionV3: This model demonstrates the highest accuracy and precision. Are there any potential drawbacks to consider despite these strong results?
InceptionResNetV2: This model shows moderate performance across all metrics. Are there any specific applications where this level of performance would be sufficient?
DenseNet121: This model has strong performance across all metrics. Do you think its architecture offers any advantages over other models?

Additional comments

Hyper-parameter Tuning: Have you explored the impact of hyper-parameter tuning on the performance of these models?
Data Augmentation: Were any data augmentation techniques employed, and if so, did they significantly impact the results?
Computational Cost: Have you considered the computational cost of training and deploying each model?
The paper addresses an important topic by leveraging advanced machine learning techniques in agriculture. The comparative approach is valuable, but the discussion section could better connect the findings to practical implications, such as how the best-performing model can be applied in real-world coffee bean sorting systems. Consider including a diagram of the model architectures to aid reader understanding

Annotated reviews are not available for download in order to protect the identity of reviewers who chose to remain anonymous.

·

Basic reporting

Author has proposed ‘Comparison of deep learning models in automatic classification of coffee bean species’, I have following comments.
- Abstract:
- Line 19: this sentence ‘In this study, considering this richness of …’ is hanging, author should fix it, make it active or passive. Subject is missing.
- It is good that author has mentioned the number of classes, however, total number of images of dataset is missing, author should add that.
- It is not wise to mention the evaluation matrix in the abstract line 24, 25, 26.
- Author should also mention the precision of all the models as well as highlight the lowest loss rate of the optimal model to make the abstract rich.
- Introduction:
- While the introduction covers the significance of coffee bean classification, it could be enhanced by clearly outlining the specific research gap the study addresses. How does it significantly advance beyond previous work?
- The introduction assumes some familiarity with deep learning for the audience. Providing a brief explanation of why the chosen architectures are relevant for this problem would strengthen the context.
- The contribution of this study is missing in introduction. It is suggested that author mentions list of contributions at the end of the introduction.
- Related work:
- This section offers an extensive literature review but could benefit from a more critical discussion. Highlighting gaps or limitations in prior studies would better position the novelty of this research.
- Comparisons to state-of-the-art methods, especially transformer-based approaches mentioned later, are mentioned but not fully explored in the context of their advantages or limitations.
- From the related work it can be also noticed that not enough recent studies in the domian of deep learning have been quoted.
- It is suggested that the author enhance that by adding more recent studies and highlight their contribution such as Enhancing soybean classification with modified inception model: A transfer learning approach; Enhanced corn seed disease classification: Leveraging MobileNetV2 with feature augmentation and transfer learning; Classification of hazelnut kernels with deep learning; Deep learning-based classification of alfalfa varieties: A comparative study using a custom leaf image dataset; Adaptability of deep learning: datasets and strategies in fruit classification.
- Materials and Methods:
- While the dataset preparation and preprocessing steps are well described, the rationale for choosing only three coffee bean types (Espresso, Kenya, and Starbucks Pike Place) is not sufficiently justified. Explain why these specific classes were selected.
- The use of transfer learning is appropriate, but there is no discussion of why these particular CNN architectures were chosen over others. Adding this reasoning would clarify their suitability for the task.
- Results:
- The results are comprehensive but lack statistical tests or significance measures to validate the differences between model performances. Including these would lend more rigor to the analysis.
- The discussion on misclassification rates is useful but could be expanded to hypothesize why certain classes (e.g., Starbucks Pike Place) are more challenging to classify.
- The results should be enhanced by adding graphs in terms of training, validation, and testing of all the models as well as confusion matrix of all the models.
- I think there should be also a comparison of models before and after using transfer learning.
- Discussion:
- The discussion effectively highlights the performance of InceptionV3 but lacks a detailed exploration of the trade-offs between computational efficiency and accuracy for the various models tested.
- Practical implications for industrial applications are mentioned but not explored in detail. Elaborating on how the findings can be integrated into real-world settings would enhance relevance.
-

Experimental design

above mentioned

Validity of the findings

above mentioned

Additional comments

Author has proposed ‘Comparison of deep learning models in automatic classification of coffee bean species’, I have following comments.
- Abstract:
- Line 19: this sentence ‘In this study, considering this richness of …’ is hanging, author should fix it, make it active or passive. Subject is missing.
- It is good that author has mentioned the number of classes, however, total number of images of dataset is missing, author should add that.
- It is not wise to mention the evaluation matrix in the abstract line 24, 25, 26.
- Author should also mention the precision of all the models as well as highlight the lowest loss rate of the optimal model to make the abstract rich.
- Introduction:
- While the introduction covers the significance of coffee bean classification, it could be enhanced by clearly outlining the specific research gap the study addresses. How does it significantly advance beyond previous work?
- The introduction assumes some familiarity with deep learning for the audience. Providing a brief explanation of why the chosen architectures are relevant for this problem would strengthen the context.
- The contribution of this study is missing in introduction. It is suggested that author mentions list of contributions at the end of the introduction.
- Related work:
- This section offers an extensive literature review but could benefit from a more critical discussion. Highlighting gaps or limitations in prior studies would better position the novelty of this research.
- Comparisons to state-of-the-art methods, especially transformer-based approaches mentioned later, are mentioned but not fully explored in the context of their advantages or limitations.
- From the related work it can be also noticed that not enough recent studies in the domian of deep learning have been quoted.
- It is suggested that the author enhance that by adding more recent studies and highlight their contribution such as Enhancing soybean classification with modified inception model: A transfer learning approach; Enhanced corn seed disease classification: Leveraging MobileNetV2 with feature augmentation and transfer learning; Classification of hazelnut kernels with deep learning; Deep learning-based classification of alfalfa varieties: A comparative study using a custom leaf image dataset; Adaptability of deep learning: datasets and strategies in fruit classification.
- Materials and Methods:
- While the dataset preparation and preprocessing steps are well described, the rationale for choosing only three coffee bean types (Espresso, Kenya, and Starbucks Pike Place) is not sufficiently justified. Explain why these specific classes were selected.
- The use of transfer learning is appropriate, but there is no discussion of why these particular CNN architectures were chosen over others. Adding this reasoning would clarify their suitability for the task.
- Results:
- The results are comprehensive but lack statistical tests or significance measures to validate the differences between model performances. Including these would lend more rigor to the analysis.
- The discussion on misclassification rates is useful but could be expanded to hypothesize why certain classes (e.g., Starbucks Pike Place) are more challenging to classify.
- The results should be enhanced by adding graphs in terms of training, validation, and testing of all the models as well as confusion matrix of all the models.
- I think there should be also a comparison of models before and after using transfer learning.
- Discussion:
- The discussion effectively highlights the performance of InceptionV3 but lacks a detailed exploration of the trade-offs between computational efficiency and accuracy for the various models tested.
- Practical implications for industrial applications are mentioned but not explored in detail. Elaborating on how the findings can be integrated into real-world settings would enhance relevance.
-

---

## Round 0.2 · accepted · Accept

· Academic Editor

Accept

All the suggestions made by the reviewers have been addressed.

Congratulations.

Reviewer 1 ·

Basic reporting

There is a noticeable improvement in the updated manuscript. The author has successfully resolved the earlier issues with the introduction and abstract. The research is now introduced more clearly and engagingly, and the abstract now appropriately conveys the study's results and breadth. Additionally, adding more recent publications to the references shows a deep engagement with current research and reinforces the paper's context. Now, the text is much closer to being published-ready. I advise doing one last, thorough check to make sure there are no last typos or grammatical issues.

Experimental design

The CNN architecture in Figure 1 is now more understandable and instructive. Figure 2, which shows coffee beans, has much improved picture quality, which increases the effectiveness of the visual depiction. The architectural schematic in Figure 4 has also been enhanced and is now easier to read. In addition, the tables have been updated to include comparisons with more sophisticated deep learning methods, offering a more thorough examination of the outcomes. The text is now much stronger as a result of these modifications. Before publishing, I advise doing a last check to make sure everything is consistent and to fix any last-minute problems.

Validity of the findings

The author has made revisions to address the previous concerns regarding the experimental setup and novelty. While the initial submission relied heavily on comparisons with pre-trained models, the revised manuscript now presents a more robust experimental design. It appears the author has incorporate [ novel model architecture, training and validation details, dataset].

·

Basic reporting

no comments

Experimental design

no comments

Validity of the findings

no comments

Additional comments

author has incorporated all the comments and the quality of the paper has been improved